# Automatic 1D ^1^H NMR Metabolite Quantification for Bioreactor Monitoring

**DOI:** 10.3390/metabo11030157

**Published:** 2021-03-09

**Authors:** Roy Chih Chung Wang, David A. Campbell, James R. Green, Miroslava Čuperlović-Culf

**Affiliations:** 1School of Mathematics and Statistics, Carleton University, Ottawa, ON K1S 5B6, Canada; roychihchungwang@sce.carleton.ca (R.C.C.W.); davecampbell@math.carleton.ca (D.A.C.); 2Department of Systems and Computer Engineering, Carleton University, Ottawa, ON K1S 5B6, Canada; jrgreen@sce.carleton.ca; 3Digital Technologies Research Center, National Research Council, Ottawa, ON K1A 0R6, Canada; 4Department of Cellular and Molecular Medicine, University of Ottawa, Ottawa, ON K1N 6N5, Canada

**Keywords:** NMR, metabolomics, bioreactors, metabolite quantification, quantitative NMR, biomanufacturing, bioprocessing

## Abstract

High-throughput metabolomics can be used to optimize cell growth for enhanced production or for monitoring cell health in bioreactors. It has applications in cell and gene therapies, vaccines, biologics, and bioprocessing. NMR metabolomics is a method that allows for fast and reliable experimentation, requires only minimal sample preparation, and can be set up to take online measurements of cell media for bioreactor monitoring. This type of application requires a fully automated metabolite quantification method that can be linked with high-throughput measurements. In this review, we discuss the quantifier requirements in this type of application, the existing methods for NMR metabolomics quantification, and the performance of three existing quantifiers in the context of NMR metabolomics for bioreactor monitoring.

## 1. Introduction

Metabolites are both downstream products and regulators of the majority of biological processes. The ability to quantify them in a nondestructive manner from a test sample offers us an opportunity to use them as a proxy to study or monitor these processes in a variety of biological systems, including live cell cultures. If automated, this would enable in vivo repeated inference of cell phenotype and its gene or protein interactions at regular time intervals, possibly in a high-throughput setup. Some high-throughput metabolomics applications include drug discovery [1,2,3], toxicology [4], biomass processing [5], and vaccine production in bioreactors [6,7].

Nuclear magnetic resonance (NMR) spectroscopy and mass spectrometry (MS) are presently the most popular methods to analyze metabolites. Although MS is more sensitive and can, with application of different MS approaches, detect more metabolites than NMR, NMR is attractive for quantitative metabolomics because NMR experiments can be performed easily and quickly without complicated sample preparation procedures. Following investment in an NMR instrument, experiments are inexpensive, and highly reproducible. MS experiments require more involved preparations, extensive quality control, and are destructive to the test sample. The nondestructive aspect of NMR makes it interesting as a first step in further, perhaps more time-consuming or expensive analysis as well as possible analysis of live cells. In bioreactor applications, NMR metabolomics is uniquely attractive for continual media monitoring, which can provide extensive data for condition optimization. Flow NMR probes allow continual sample measurement with no need for NMR sample tubes, or extensive sample preparation such as spectrometer shimming as only the sample solution itself is exchanged. Quantitative NMR metabolomics can be envisioned as an additional sensor for monitoring the molecules involved in the reactor process.

The most popular approach in NMR metabolomics, including quantitative NMR metabolomics, is the 1D ${}^{1}$H NMR. Other nuclei, such as ${}^{13}$C, ${}^{15}$N, or ${}^{31}$P can also be used in natural abundance or in isotopically labeled samples. The physics behind the 1D ${}^{1}$H NMR relate the concentration of the metabolites’ signals in a straightforward manner [8], and an experiment can be performed in a matter of minutes. We refer the reader to the review by Emwas et al. [9] for more information on the different NMR methods and its comparison to MS for metabolomics.

The 1D NMR has a relatively narrow spectral range, and multidimensional NMR can better identify a large number of metabolites in a test sample. However, multidimensional NMR experiments generally take significantly longer to perform, and the concentration of each metabolite in the test sample could have different proportionality constants to the signal volume [8,9,10]. Recent literature, such as [10], provide strategies and protocols to mitigate the different proportionality constants and slow acquisition time issues, and quantification from 2D NMR seems attainable under certain conditions. Still, 1D NMR experiments remain a preferred method for most metabolomics applications due to its fast measurement time, which is a desirable characteristic in a monitoring application. There are also more 1D ${}^{1}$H NMR metabolite standards in public databases than any other types of NMR, which is useful for the metabolite assignment and quantification task.

Presently, most metabolomic studies rely on manual or semimanual NMR spectra profiling, identification, or quantification by trained experts. This is because profiling many different metabolites from standard libraries remains an open pattern recognition problem. Although there are guides on this manual task (e.g., [11]) and several automated and semiautomated methods have been published [12,13,14,15], a fully automated solution remains elusive. It would also be beneficial if such an algorithm could provide some form of uncertainty quantification to better inform downstream analysis algorithms. In addition, existing literature requires specific experimental conditions or experimental designs for accurate profiling or quantification of 1D ${}^{1}$H NMR spectra; for example, a set of protocols were identified for plant metabolomics in [16], blood profiles in [17], celebrospinal fluid in [18], urine in [19], and experiment design methods were proposed for the bioreactor setting in [20].

Most existing quantitative NMR literature focus on biofluids that are relevant for clinical applications of NMR metabolomics. Many such applications have test samples that arise from controlled environments, which are similar to the standards from public libraries [9,21,22]. Conversely, bioreactors, generally defined as vessels where biological reaction or change is taking place, can involve enzymes (e.g., for cell-free bioprocessing), microorganisms (e.g., for environmental remediation or fermentation), animal cells (e.g., for production of biologics or vaccines), plant cells (e.g., for production of bioactive compounds), and tissues (e.g., for cell therapies). To provide a reliable yield in these types of bioreactor applications, it is crucial to maintain a suitable environment for the desired biological reaction to take place. This is only possible with sufficient information about the cellular environment at different time stamps of the reactor process.

Most existing quantifiers also focus on applications where the metabolites have mostly reached homeostasis, and that the metabolic profiles differ mostly in concentration but not composition, i.e., no new metabolites are being introduced between experiments. In the case where multiple experiments were conducted at different times during a particular run of a bioreactor, some of the experiment samples could contain different types of byproduct metabolites that are not present in the other samples. Therefore, methodologies for fast, high-throughput quantitative metabolomics of bioreactors require quantifiers that are robust to the challenges in this setting. We discuss these issues in Section 3.

There are a few recent reviews that touch on quantitative NMR (e.g., [9,21,22]). However, to the best of our knowledge, benchmark or performance analysis for existing quantitative NMR approaches applied to the bioreactor setting is lacking. We first review a popular 1D ${}^{1}$H NMR signal model, followed by some challenges for quantifying metabolite concentrations. We then briefly review some existing NMR quantifiers before reporting our benchmark results. Our motivation for the benchmark section is to provide insight into how some existing NMR quantifiers [12,14,15] perform against our benchmark sample. These quantifiers were chosen because they are publicly accessible and are still being maintained. We used Dulbecco’s modified Eagle’s medium (DMEM), a cell growth medium, as our benchmark biological sample. The experiment associated with our benchmark data was conducted using a 600 megahertz (MHz) spectrometer. This cell medium was chosen because of the presence of glucose and a number of other amino acids that are commonly found in mammalian cell bioreactor applications. We conclude with some potential avenues of exploration for future quantifiers. The benchmark NMR data is included in the Appendix A.

## 2. Preliminaries: 1D ${}^{1}$H NMR

Modern NMR spectrometers operate on the principles of Fourier NMR theory [23,24]. The collected data from a 1D ${}^{1}$H NMR experiment can be approximated by the free-induction decay (FID) signal model:(1)$$s\left(t\right)=\sum_{l}\alpha_{l}e^{i\left(\Omega_{l}t-\beta_{l}\right)}e^{-\lambda_{l}t},$$
where *i* is the imaginary number, $\Omega_{l}$, $\alpha_{l}$, and $\lambda_{l}$ are all positive real numbers, and $\beta_{l}$ is an angle in radians. This is a weighted sum of complex-valued exponential components, and we shall denote by *l* the resonance component index. The frequency term $\Omega_{l}$ captures the energy difference of the excited protons in a molecule, $\alpha_{l}$ acts as a measure of component intensity, and $\beta_{l}$ quantifies a phase offset from the other components. The decay term $\lambda_{l}$ is related to the transverse relaxation time constant T2, which is a characteristic of the molecule. In practice, many of the components have similar frequency values, and such components visually appear as one unified component in the observed data spectrum due to the finite spectral resolution of the instrument. Further details on NMR theory can be found in [8,25,26], and we focus on the implications of these effects on the observed data in this section.

The sinusoidal components in the FID signal can have frequencies that are too high for a human to visually appreciate, so NMR spectroscopists usually work with the Fourier transform of the FID signal. The Fourier transform of the FID is
(2)$$\begin{array}{ccc} S\left(\nu \right) & = & \sum_{l}\alpha_{l}e^{i\beta_{l}}Q_{l}\left(\nu \right), \end{array}$$
where the ν is the frequency variable in Hertz (Hz), and the complex Lorentzian is defined as
(3)$$\begin{array}{ccc} Q_{l}\left(\nu \right) & := & \frac{1}{\lambda_{l}+i\left(2\pi \nu -\Omega_{l}\right)}. \end{array}$$

The NMR literature often work with the real and imaginary parts of the complex Lorentzian. The real part is called the absorption Lorentzian and is given by
(4)$$\begin{array}{ccc} \mathrm{real}\left\{Q_{l}\left(\nu \right)\right\} & := & \frac{\lambda_{l}}{\lambda_{l}^{2}+{\left(2\pi \nu -\Omega_{l}\right)}^{2}}. \end{array}$$

This function is strictly positive, unimodal, and has heavier tails than a Gaussian function. The width and tail characteristics of this function are both controlled by $\lambda_{l}$. A heavier molecule usually has a smaller T2 relaxation term than a lighter molecule, which corresponds to having a Lorentzian with greater widths. Take note that both the α and λ parameters affect the visual height of the absorption Lorentzian. This can be inferred by substituting the frequency ν by its resonance (peak) frequency $\frac{\Omega_{l}}{2\pi }$ in Equation (Equation 4), which gives the peak amplitude $\frac{1}{\lambda_{l}}$ for the absorption Lorentzian. Figure 1 illustrates the effect of λ on the visual height and tail characteristics of the absoption Lorentzian. Here, the two Lorentzians share the same the intensity α value.

The imaginary part is called the dispersion Lorentzian and is given by
$$\begin{array}{cc} \mathrm{imag} & \left\{Q_{l}\left(\nu \right)\right\}:=\frac{2\pi \nu -\Omega_{l}}{\lambda_{l}^{2}+{\left(2\pi \nu -\Omega_{l}\right)}^{2}}. \end{array}$$

From these definitions and Equation (Equation 2), it is clear the phase $\beta_{l}$ facilitates a mixing role for how much absorption and dispersion Lorentzian is present in the real and imaginary parts of the FID spectrum *S*.

### 2.1. Intensity

For 1D ${}^{1}$H NMR spectroscopy, the intensity parameter $\alpha_{l}$ is directly proportional to the number of hydrogens that is associated with the resonance component *l* [8,25,26]. The absorption Lorentzian in Equation (Equation 4) is often used by NMR spectroscopists because it offers a way to estimate $\alpha_{l}$ given $\beta_{l}$, without having to fit the $\Omega_{l}$ and $\lambda_{l}$ parameters. The definite integral of the absorption Lorentzian over the interval $\left[c_{1},\;c_{2}\right]$ is given by
(5)$$\begin{array}{ccc} \int_{c_{1}}^{c_{2}}\frac{\lambda_{l}}{\lambda_{l}^{2}+{\left(2\pi \nu -\Omega_{l}\right)}^{2}}d\nu & = & \frac{1}{2\pi }\left(\tan^{-1}\left(\frac{\Omega_{l}-2\pi c_{1}}{\lambda_{l}}\right)-\tan^{-1}\left(\frac{\Omega_{l}-2\pi c_{2}}{\lambda_{l}}\right)\right). \end{array}$$

Consider a signal $\tilde{S}$ that is exactly like *S*, but has each of its phase parameters, $\beta_{l}$, set to zero for all resonance indices *l*. The real part of Equation (Equation 2) in this setting becomes
$$\begin{array}{ccc} \mathrm{real}\left\{\tilde{S}\left(\nu \right)\right\} & = & \sum_{l}\frac{\alpha_{l}\lambda_{l}}{\lambda_{l}^{2}+{\left(2\pi \nu -\Omega_{l}\right)}^{2}}. \end{array}$$

When $c_{1}\to -\infty$ and $c_{2}\to \infty$ in Equation (Equation 5), i.e., when integrating over the entire domain, we have
$$\begin{array}{ccc} \sum_{l}\int_{-\infty }^{\infty }\frac{\alpha_{l}\lambda_{l}}{\lambda_{l}^{2}+{\left(2\pi \nu -\Omega_{l}\right)}^{2}}d\nu & = & \sum_{l}\frac{\alpha_{l}}{2}. \end{array}$$

This implies that one can get an approximation of any $\alpha_{l}$ that has a resonance frequency $\Omega_{l}$ far from any other resonance frequency by integrating $\mathrm{real}\{\tilde{S}\left(\nu \right)\}$ over a domain where there are no other resonance components. In practice, one needs to estimate $\tilde{S}$ given time series data of the FID signal *s* (see Equation (Equation 1)). One then assumes the resonance index *l* of interest has a resonance frequency $\Omega_{l}$ that is far from the frequencies of the other resonance components, and integrate $\mathrm{real}(\tilde{S})$ near $\Omega_{l}$ to get an estimate for $\alpha_{l}$.

One can think of $\tilde{S}$ as a canonical representation that could be estimated from the collected data of the time series, *s*, before performing metabolite assignment or quantification procedures on this estimated canonical representation. This reason motivates spectroscopists to develop methods to estimate the zero-phased spectrum $\tilde{S}$ given the spectrum *S*. In reality, one can only approximate the Fourier transform *S* from the finite samples of the data-domain data *s*, since one can only compute the discrete-time Fourier transform or the discrete Fourier transform from the data samples of the FID signal *s*. The estimated $\tilde{S}$ is commonly referred to as the phase-corrected, phased, or auto-phased spectrum. We shall discuss the estimation of $\tilde{S}$ in Section 3.5.

### 2.2. Metabolite NMR

The unique peak patterns for each metabolite in the test sample is a manifestation of its spin interactions with an external magnetic field and radio-frequency pulse excitation. Some patterns are attributed to parameters that are intrinsic to a molecule, with chemical shift and *J*-coupling constants being the most relevant in the final NMR spectrum. The chemical shift parameter can also be affected by various external factors such as the potential of hydrogen (pH) of the sample or the presence of other metabolites. The frequency units in NMR spectroscopy are usually ppm or Hz; chemical shifts are expressed in ppm, and *J*-coupling constants are expressed in Hz.

Informally speaking, these parameters are a characterization of the location of the nucleus of the ${}^{1}$H spin and its electronic environment. They affect the resonance frequency position $\Omega_{l}$ and intensity $\alpha_{l}$ of that resonance component. There are existing studies that estimate these parameters from empirical data [27,28], as well as works that simulate the NMR spectrum given these parameters [29]. See [30] for a brief guide on how these parameters can help with identifying metabolites, and [31] for a survey of various methodologies to estimate them.

## 3. Challenges for Automatic Quantification for Bioreactor Monitoring

We summarize in Table 1 some differences between NMR experiments collected from cell cultures and biofluids. Continual monitoring refers to the collection of samples and NMR experimentation in short time intervals, and is only possible in the bioreactor application of cultivating cell cultures. The number of samples for a generic bioreactor application is based on the assumption of having a continual monitoring setup where measurements are carried out every 15 min over a 20-day bioreactor run. This would result in 1920 NMR experiments. On the other hand, biofluid analysis requires either animal models or patients samples, which makes it difficult to collect a large numbers of samples.

The challenges of quantifying a generic 1D ${}^{1}$H NMR cell culture experiment are: (1) many metabolites have overlapping peaks due to similar resonance frequencies, (2) low sensitivity of the NMR experiment can lead to unresolved metabolites of interest, (3) the possibility of unknown contaminants in the test sample, (4) differences in experimental conditions (especially for online monitoring) can cause a discrepancy between the catalogued spectra from NMR databases and the observed spectrum in experiment at hand for a given metabolite, and (5) the presence of the phase parameter $\beta_{l}$.

Specifically to the bioreactor application, the main issue is in the large number of experiments that need to be performed and quantified quickly, which makes methods that require extensive sample processing and application of 2D NMR difficult. An automated quantifier is required because it is impractical to manually input different parameters for each experiment in a continual monitoring setup. In this setting, an automated NMR quantifier needs to be robust to unpredictable small changes in metabolite peak positions (osmolality and pH), the presence of a strong solvent (water) signal, the possible presence of larger particles (e.g., exosomes) if a filtering or centrifugation step is not performed, and a short data acquisition time.

### 3.1. Peak Overlap

The spectral window of 1D ${}^{1}$H NMR spectroscopy is relatively narrow for spectroscopy, and it is likely that some of the metabolites in a sample will have their most prominent resonance frequencies close to each other. This visually appears as overlapping peaks in the data spectrum. Such an example is shown in Figure 2 for l-Leucine, l-Isoleucine, l-Valine, and a bioreactor media that contains these three amino acids.

To simplify the pattern matching, some Fourier-based quantifiers select frequency ranges that are associated with only the most prominent peaks for each metabolite, as opposed to modeling the peak pattern for the entire frequency range. For manual quantification, one typically looks at the uncluttered frequency ranges that is specific to the metabolite of interest. This approach has poor performance when there are spectral overlaps from the different types of metabolites in the sample. There may also be ambiguous identifications of metabolites that have similar selected frequency ranges. For this reason, some form of uncertainty quantification over the concentration estimates is desirable.

### 3.2. Experimental Sensitivity

Low concentrations of metabolites such that its signal is barely registering in the NMR experiment data can lead to an unreliable concentration estimate. Relative ratio analysis of the low concentration metabolite, while in the presence of a higher concentration metabolite, may result in a significant error level. This is problematic in samples that have a high concentration dynamic range. In the bioreactor setting, both high- and low-level metabolites can affect cell growth. Furthermore, the concentration of metabolites can change dramatically over time. For example, a significant concentration difference between toxic byproducts and nutrients is usually of interest to bioreactor metabolomic studies.

### 3.3. Contaminants

A quantification scheme that compares the observed data against a database of catalogued metabolites can have performance issues when the observed peaks do not line up with the catalogued metabolite representations. This might occur when experimental conditions differ from the conditions used to produce the database and leads to misidentification of metabolites. If the user cannot constrain the quantifier to a preset target of metabolites, this could lead to a large false positive identification error. A robust quantifier needs to be able to achieve good sensitivity and specificity as measures of the misidentification of compounds. Quantifiers that only use a limited spectral region (i.e., only selected peaks) as a way to represent a metabolite are more susceptible to misidentification. Furthermore, the observed peaks in frequency regions that do not correspond to a catalogued metabolite are treated as a contaminant compound. This complicates contaminant discovery and analysis studies since parts of noncontaminant metabolites that failed to align with the catalogued spectra are treated as signals from potential contaminant metabolites.

An example is the internal reference compound sodium trimethylsilylpropanesulfonate (DSS), which has a prominent peak at 0 ppm, but also has secondary peaks around 0.6, 1.6, and 2.9 ppm. If a quantification scheme does not use the latter three regions as its internal representation of DSS, then DSS’ spectrum between 0.6 and 2.9 ppm in the data will appear to be a contaminant. This is especially troublesome when quantifying metabolites that have a much lower concentration than DSS and have small multiplet peaks around 0.6, 1.6, and 2.9 ppm. For example, l-Leucine also has a small multiplet pattern around 1.7 ppm, and a quantifier algorithm that only considers the 1.7 ppm range to quantify l-Leucine could have performance issues when both DSS and l-Leucine are present in the sample. DSS could possibly also react with certain metabolites [32]. Although more inert reference compounds have been proposed [32], most of the entries in publicly available NMR metabolomics databases use DSS as the reference. Other popular NMR references include trimethylsilylpropanoic acid (TMSP or TSP) and tetramethylsilane (TMS), and the user has to consider possible differences in relative peak positions due to change in the 0 ppm reference.

### 3.4. Different Experimental Conditions

Our main focus in this paper is on mammalian cell bioreactors, which are often used for vaccine production and gene therapy applications. Aqueous media is used in these applications, and the large concentration of the water solvent presents a challenge for NMR experiment. Water-drying of samples is possible in a bioreactor setup, but impractical if a large number of samples are required for the NMR experiment. Additionally, drying is impossible if NMR experiments are performed using flux probe systems that directly collect and measure samples from the reactor without any preprocessing steps. In this case, NMR experiment protocol-based solvent suppression techniques can attenuate the large resonance signal from the solvent. However, such a technique could also distort the resonance frequencies near the solvent’s resonance frequency [33].

The resonance frequencies and intensities that make up a metabolite’s spectral profile can be affected by a sample’s pH and osmolality. Although the sample is usually buffered to 7–7.4 pH, this monitoring is difficult to enforce in an online monitoring application. Solution osmolality depends on the type and amount of all metabolites that are present in the test sample. In [34], it was found that resonance frequency shift due to different concentration makeup could be predicted for a specific mixture of 60 metabolites that are common in urine, provided that the concentration of a few select key metabolites are estimated correctly in a preprocessing step from the test sample. Unfortunately, this type of study is only emerging, and public databases presently do not have enough data of this type to provide a training set for constructing a data-driven peak shift prediction model.

### 3.5. Phase and Baseline

A popular preprocessing step to quantification is to autophase the data spectrum *S*. The idea is to treat the phase term $\beta_{l}$ in Equation (Equation 2) as an artifact, and try to exclude it from the preprocessed data spectrum $\tilde{S}$. The use of an autophased spectrum of the NMR experiment data $\tilde{S}$ does not require one to have knowledge of the resonance frequencies $\Omega_{l}$ before quantification.

Most autophasing algorithms assume the phase of the observed spectrum follows the affine model
(6)$$\beta \left(\nu \right)=c_{0}+c_{1}\nu ,$$
and the phase terms $\beta_{l}$ from Equation (Equation 2) are assumed to satisfy
(7)$$\beta_{l}\approx c_{0}+c_{1}\frac{\Omega_{l}}{2\pi }.$$

The autophasing algorithm first estimates the parameters $c_{0}$ and $c_{1}$ from the data spectrum; then, it modifies the data such that the $\beta_{l}$ terms in the modified data’s spectrum $\tilde{S}$ are all close to zero. Most algorithms do this by multiplying the data spectrum *S* by the negative affine phase model, i.e.,
(8)$$\tilde{S}\left(\nu \right)=S\left(\nu \right)e^{-\beta (\nu )}.$$

If Equation (Equation 7) is a good approximation for each phase term $\beta_{l}$, then the phase terms will be close to zero in the transformed spectrum $\tilde{S}$. Many quantifiers take the real part of $\tilde{S}$ as a weighted sum of absorption Lorentzian components, therefore avoiding $\beta_{l}$ in subsequent steps in their quantification procedure.

The disadvantage of using Equation (Equation 8) is that all off-resonance frequencies (i.e., any $\nu \neq \frac{\Omega }{2\pi }$) are subjected to a frequency-dependent phase distortion by −β(ν). These distortions may unpredictably manifest as a negative peak in the real part of the transformed spectrum, especially at frequency locations that have significant tail contributions from nearby Lorentzian components. Figure 3 is an example of a negative peak around the water solvent peak (4.7 ppm). Despite this issue, autophasing is a simple procedure that is widely used in the community. We refer the reader to [35] for a comparison of existing autophasing algorithms.

In practice, one performs either the discrete-time Fourier transform or the discrete Fourier transform to the sampled FID time-series data to approximate $\tilde{S}$. Present-day NMR spectrometer also exhibit some time delay between the end of the excitation radio pulse sequence and the collection of data. The analog and digital electronic filters in the spectrometer might have problematic nonlinear phase characteristics. The magnetic field may also be nonuniform across the sample. These and others factors (e.g., see [36]) all contribute toward a discrepancy between the spectrum of the collected time series data and the Lorentzian model that was discussed in Section 2. In the NMR literature, these effects are referred to as baseline distortion. The issues from hardware and Fourier transform approximation are discussed in detail in [37], as well as possible methods to mitigate them through experiment protocol.

Despite recent advancements in spectrometer technology and having better NMR protocol design knowledge, the use of autophasing as a preprocessing step would still introduce some distortion to the data, sometimes significantly. Therefore, many quantifiers employ the use of baseline compensation algorithms (e.g., [38,39,40,41]) after autophasing so that the data can be well-approximated by the Lorentzian model.

### 3.6. Model Parameterization

As we discussed in Section 3.4, the peaks in the observed spectra of a sample is likely to differ from a catalogued spectra due to different experimental conditions and sample osmolality. An automated approach to quantification requires one to either predict the perturbation behavior, or to solve for nuisance variables that parameterize the peak shifts from the data.

The first approach poses a challenge since osmolaity is difficult to analytically quantify in our setting. However, the data-driven chemical shift prediction mapping reported in [15] could render this approach relevant for the future generation of quantifiers. The second approach can get computationally intensive as the number of metabolites to be quantified increases, since the number of nuisance parameters are related to the number of resonance components in the data spectrum. This approach is not scalable unless approximations to the data generation model are made. For this reason, most quantifiers use only a select number of peaks to characterize a metabolite. Autophasing is usually done as a preprocessing step to avoid having to estimate the phase parameters jointly with the metabolite concentrations.

## 4. Existing Quantifiers

Existing quantifiers that use the FID signal model have the advantage that the FID model is a generative model for the data, i.e., no Fourier transform is used in the data model. This makes it possible to directly quantify uncertainty under a Bayesian inference framework. It does not have issues due to unpredictable baseline nor phase distortion that arise from Fourier transform approximation and autophasing. The disadvantage is that all of the FID parameters and the metabolite concentrations must be accurately estimated.

Working in the time domain involves fitting the model (Equation (Equation 1)) to the observed time series (the FID data), but the overlapping set of an unknown number of resonance frequencies places significant computational strain on the numerical optimization algorithm used. Due to the number of FID variables involved with just a single metabolite, it is difficult to scale up this approach to handle the number of metabolites and experiments that are needed in a bioreactor monitoring application. Quantifiers that use the Fourier transformed FID signal model tend to require less computational resources, since the resonance frequencies are easier resolved in the Fourier domain without requiring accurate estimates of the other FID parameters. The quantifiers that we benchmark in Section 5 all use this type of signal model.

As discussed in Section 3.6, most of the Fourier domain quantification schemes use some autophase algorithm as a preprocessing step to reduce the computational burden. However, these preprocessing steps could have issues when the signal-to-noise ratio of the NMR experiment is low. Since time-domain NMR quantification approaches typically avoid such preprocessing steps, they could perform better on experiments that are collected in low signal-to-noise ratio environments. Such a time-domain quantifier for quantifying a low number of compounds was reported by Matviychuk et al. [42].

Some recent quantifiers place constraints on the FID parameters. E-RANSYS (Extractive ratio analysis NMR spectroscopy) [43] and AQuA (Automated quantification algorithm) [44] work by constraining the relative ratio of the amplitude of peaks of each metabolite in its internal library. ASICS (Automatic statistical identification in complex spectra) [15] fits a deformation mapping of the peak positions to account for peak shifting. However, since the resonance frequency shiftdue to experimental conditions is hard to predict (see Section 3.4), such a deformation mapping approach to predict the peak shifts should require a sizeable amount of data, sampled at various concentrations and chemical environment conditions. The promising report [34] used a few thousand NMR experiments sampled at different concentrations to construct a predictive mapping for the chemical shift of selected metabolites that are common in urine. Peak alignment approaches such as [45] can be used prior to the quantification to compensate for peak shifts. Due to user-induced bias in the adjustment of peak positions, this approach can lead to assignment and overestimation errors.

Some quantifiers require input parameters that needs to be manually determined from the data of each NMR experiment to be quantified. The BATMAN (Bayesian automated metabolite analyzer for NMR) quantifier [46] requires the user to specify various peak location and shift values of the targeted metabolites. This is a time-intensive and error-prone task, especially when the required information must be determined individually for each peak of each metabolite. Differences in spectral resolution of the instrument, peak overlap, and peak shift can lead to errors in this approach even after the extensive manual user-led input step.

Although there are a few NMR quantifiers in the literature, some are no longer maintained, e.g., BATMAN. Many require proprietary software to run, e.g., E-RANSYS, AQuA, and the quantifier by Filntisi et al. [47]. In this review, we focus on the Bayesil [12], the ASICS [15], and the rDolphin [14] quantifiers. These quantifiers employ third-party autophasing and baseline correction as preprocessing steps, and are currently accessible to the public for running quantification jobs. They do not require extensive manual input of metabolite-specific instructions, making them appropriate for automated quantification.

### 4.1. Bayesil

The Bayesil quantifier [12] optimizes nuisance variables that perturb the peak positions of its internally catalogued metabolite representations. Each catalogued metabolite stores frequency region information that are called clusters. The library was constructed using the spectra and metadata from the Human Metabolome Database (HMDB) at www.hmdb.ca (8 March 2021). This quantifier assumes the peaks in each cluster all shift by the same amount and in the same direction. A common shift variable with a window of +/−0.025 ppm is assigned to each region. The algorithm uses a cost function to fit the reconstructed Lorentzian spectrum against the autophased data spectrum $\tilde{S}$. The cost function is the sum-of-squares difference between the two spectra with a penalty term based on discrepancy of the derivatives. Further details are available in the Supplementary Materials of [12]. The frequency shift nuisance variables for each cluster in each metabolite is jointly estimated with the concentration of each metabolite.

The cost function is treated as the energy in a Gibbs distribution, and a sequential Monte Carlo (SMC) algorithm is used to iteratively refine a finite population of candidate solutions of the shift and concentration variables. The population mode of the last iteration for each variable is taken as the solution shift and concentration values. Although technically a probabilistic inference problem, this approach to obtaining a point estimate of the variables is similar in spirit to a heuristic-search optimization algorithm, except that the algorithm strives to fit the population histogram to the Gibbs distribution. The confidence and detection threshold scores for the estimated concentrations are based on the signal-to-noise ratio of the NMR experiment and are determined by the quantifier from the metadata associated with the number of scans used in the experiment.

As discussed in Section 3.6, jointly optimizing the concentration and the shift values is a high-dimensional nonconvex problem. In addition to the use of SMC as an optimization search strategy, Bayesil utilizes existing probabilistic inference methods for efficient computation over factor graphs, which requires the variables to be placed in interdependent groups. To construct factor graphs, Bayesil segments the autophased spectrum $\tilde{S}$ into a frequency partition. Each frequency range in this partition must have the same clusters from the same metabolites.

For example, consider the scenario where there is only one cluster of peaks from metabolite A and also one cluster of peaks from metabolite B in the 1.5–3 ppm range of the NMR experiment, and that no other metabolite peaks are present in that range. Suppose the cluster from metabolite A occupies 2–3 ppm and the cluster from metabolite B occupies 1.5–2.3 ppm. The frequency partition over 1.5–3 ppm used to construct the factor graph would be the ranges 1.5–2 ppm, 2–2.3 ppm, and 2.3–3 ppm. The use of such a partition scheme assumes the user specifies all the metabolites that are present in the data. This is because the spectrum of unaccounted metabolites might invalidate the requirements of a factor graph, which might impact the quantifier’s performance.

Bayesil is a closed-source software, but is available for public use at www.bayesil.ca (accessed on 8 March 2021). The user can choose a custom set of metabolites to be used, and the data should be acquired by a 500 or 600 MHz spectrometer. Bayesil can provide the absolute concentration of metabolites if the user provides the absolute concentration of the experiment reference compound. The current version of Bayesil accepts either DSS or TSP as the reference compound. The user can select a list of metabolites from their library for a targeted quantification.

### 4.2. ASICS

The ASICS quantifier [15] used custom NMR experiments to build its internal library of at least 175 catalogued metabolites. The first version used only one 1D ${}^{1}$H NMR experiment per catalogued metabolite, but this was expanded to using multiple experiments in the latest version. Whereas Bayesil estimates shift variables for the regions in each catalogued metabolite at run-time, ASICS estimates a monotone function for each catalogued metabolite at run-time. The monotone function for a metabolite is then applied to the catalogued spectrum of that metabolite. One can think of this function as a distortion mapping that models the perturbation due to variable experimental conditions.

The weighted sum of all distorted catalogued spectra is then fitted against the data spectrum using a least absolute shrinkage and selection operator (LASSO) estimator, with concentration variables as the weights. The procedure of the sparse estimation of the weights and the monotone functions are performed in multiple stages of refinements, and a statistically motivated variation of LASSO is used to solve for the concentration variables in the final stage. The ASICS publication [15] stated that the final output of the weights are to be interpreted as a relative concentration to the most abundant compound in the test sample, but we do not observe any estimated compounds with a relative concentration of one in our benchmarks. We speculate that either ASICS automatically removes the concentration of the solvent from the reported results, or that the current version of ASICS no longer normalizes to the most abundant compound.

Unlike Bayesil, ASICS does not allow the user to select a custom list of metabolites from its library for a targeted quantification. In other words, ASICS always quantifies with the assumption that all its catalogued metabolites could be in the test sample. This is their motivation for using their sparsity-promoting LASSO estimator. They also perform a preprocessing step that removes a catalogued metabolite type from the quantification job if no significant spectral activity is observed in the data spectrum at the frequency regions associated with that catalogued metabolite. The current version of ASICS is part of the Bioconductor family of libraries for the R programming language. The library contains mostly binary files, and source code of the core methods is not provided.

### 4.3. Dolphin

The Dolphin quantifier [13] was developed for MATLAB. The rDolphin quantifier [14] is a variant of this quantifier that was developed using the R programming language. The idea behind Dolphin is to use 2D NMR data to help with quantification. For a metabolite to be quantified, it first needs to be detected by peak matching against a catalogued list of metabolites using 1D data as well as multiplet matching using 2D data. Once identified, Dolphin searches the spectral neighborhood for assigning peaks and multiplets using both 1D and 2D data. The quantification step follows, and is based on the constrained total line-shape fitting method of estimating peak parameters given multiplets [48]. The line-shape fitting algorithm fits a combination of Gaussian and Lorentzian functions to the multiplets, and the fit variables of these combination functions are subject to constraints that are based on its internal library of catalogued metabolite peaks and multiplet information. Dolphin quantifies a detected metabolite by the area under the fitted Lorentzian–Gaussian function that corresponds to the library-defined region for that metabolite.

In rDolphin, users could specify their own metabolite peaks, multiplet, chemical shift, *J*-coupling values, and allowed shift tolerance information in a custom profile file. If there are multiple frequency regions associated with a metabolite in the profile file, rDolphin returns a quantification result for every region. Unfortunately, we are unaware of a detailed peer-reviewed publication about the internal workings for rDolphin. The rDolphin software is open-source, so interested users can explore the code for this approach.

## 5. Benchmark Results

In this section, we benchmark a 1D ${}^{1}$H NMR experiment to the Bayesil, ASICS, and rDolphin quantifiers. The experiment sample is a commercial Dulbecco’s modified Eagle’s complete medium (DMEM) cell growth medium diluted at 80%. We believe this NMR experiment data is appropriate for testing performance of existing quantification methods because the concentrations of the metabolites in this sample are known from the product specification, and it is a commonly used cell medium for mammalian cell bioreactor applications. DMEM has relatively high concentration of glucose when compared with the other ingredients, and this large dynamic range is appropriate for methodology testing. NMR metabolomics measurements were performed using fast 1D experimentation and minimal sample preparation amenable to continual bioreactor monitoring application. DSS was used as the reference compound, and the solvent is deuterated water (D2O). A Bruker 600 MHz spectrometer was used in this experiment with spectral width of 10 ppm. The pulse sequence setting noesygppr1d was used. Table 2 contains a list of nonsalt compounds taken from the DMEM product specification. We have provided the NMR data for this experiment in the Appendix A, and more experiment-related settings can be found it its metadata.

ASICS and rDolphin return concentration estimates relative to some internal quantity [14,15], and Bayesil returns concentration estimates relative to a user-specified concentration of the reference compound, DSS [12]. The quantifiers did not return the same set of detected metabolites. In order to compare the results across the quantifiers, we compared each concentration estimate to the concentration estimates of both d-Glucose and l-Leucine. These two metabolites were chosen as concentration references because they were present in the concentration estimates of all three quantifiers benchmarked, except ASICS estimated d-Glucose-6-Phosphate instead of d-Glucose. l-Leucine and d-Glucose are both typically present in bioreactor metabolomic studies as a major energy molecule and an essential amino acid, respectively. They also have different absolute concentrations in the product specification (Table 2), and we can infer whether the different estimation bias caused by different amounts of the compounds in the test sample is severe enough to significantly change the computed relative concentration of the estimates.

In each row of the tables below, the relative concentration of a compound *A* with respect to d-Glucose (column *RCG*) and with respect to l-Leucine (column *RCL*) is reported. *RCG* is given by
(9)$$\begin{array}{ccc} \mathrm{RCG}\;\mathrm{of}\;\mathrm{compound}\;A & := & \frac{\mathrm{concentration}\;\mathrm{estimate}\;\mathrm{for}\;\mathrm{compound}\;A}{\mathrm{concentration}\;\mathrm{estimate}\;\mathrm{for}\;\mathrm{D-Glucose}}. \end{array}$$

A similar definition is used for *RCL*.

The relative error between a quantifier’s RCG estimate and the DMEM specification’s RCG is denoted by the column *REG*. It is given by
(10)$$\begin{array}{ccc} \mathrm{REG}\;\mathrm{for}\;\mathrm{compound}\;A & := & \frac{\left(\mathrm{quantifier}’s\;\mathrm{REG}\;\mathrm{for}\;A-\mathrm{specification}’s\;\mathrm{REG}\;\mathrm{for}\;A\right)}{\mathrm{specification}’s\;\mathrm{REG}\;\mathrm{for}\;A}, \end{array}$$
and the additive error *AEG* is defined as the numerator of the relative error. Note that this is a signed quantity, like the relative error. A similar definition is used for the relative and additive errors for the relative concentration of a compound with respect to l-Leucine, which are denoted by *REL* and *AEL*, respectively. These definitions of error imply that an overestimate yields a positive error and an underestimate yields a negative error.

### 5.1. Bayesil Benchmark

Bayesil allows the user to pick the targeted metabolites from its library. In most bioreactor applications for cell growth, the operator has a general knowledge of the possible metabolites to be expected since the desired growth reaction should be known. We selected the metabolites from this library that are in the DMEM specification. Not all compounds in the specification are present in the Bayesil library. The results are in Table 3.

Bayesil’s quantification result includes a comparison spectra plot between the preprocessed data spectrum (i.e., $\tilde{S}$ from Section 3.5) and the reconstructed spectrum that uses its quantified results. We exported this plot in the nmrML open data format [49] using Bayesil’s interface, and it is in the Appendix A. Figure 4 and Figure 5 show two different levels of detail of Bayesil’s plot.

### 5.2. ASICS Benchmark

Unlike Bayesil, ASICS does not allow the user to specify a list of metabolites to target. Instead, it automatically detects which of its 150+ catalogued metabolites are present in the data. We specified ASICS to exclude the water solvent peak region between 4.7–5 ppm. ASICS returned many concentration estimates for metabolites that were not in our DMEM product specification. In Table 4, we list only metabolites that match the contents of the DMEM in Table 2. The complete list of detected metabolites by ASICS are reported in the Appendix A. Note that we treated d-Glucose-6-Phosphate as d-Glucose in Table 4, because ASICS did not return an estimate for d-Glucose. We needed a stand-in for d-Glucose in order to compute our relative concentrations for comparison with the other quantifiers.

ASICS’ quantifier result includes samples of the preprocessed data spectrum and the reconstructed spectrum. Figure 6 and Figure 7 show two different levels of detail of this plot, and the data required for it are in the Appendix A.

### 5.3. rDolphin Benchmark

The rDolphin quantifier comes with three different region-of-interest (ROI) profiles: blood, fecal, and urine. Each profile specifies a set of metabolites that are common in a particular type of biofluid. In each of these profiles, multiple regions could be assigned to the same metabolite, and the quantifier could return a different concentration estimate for each of the regions. To report only one concentration estimate from these multiregion metabolites, we averaged the estimates from each of the metabolite’s regions. The concentration estimates relative to the concentration estimates of d-Glucose and l-Leucine are shown in Table 5, Table 6 and Table 7 for the blood, fecal, and urine profiles, respectively. The original output from rDolphin for each profile is in the Appendix A.

Unfortunately, rDolphin returned a not-a-number (NaN) value for one of the regions of d-Glucose when we used its fecal profile; see the Appendix A for the unformatted quantifier output. For Table 6, we excluded the region with the NaN estimate from the summation to compute the concentration estimate for d-Glucose. We were unable to find an option in rDolphin to visualize the reconstructed spectra of every metabolite on the same plot.

## 6. Discussion

The goal of using quantitative NMR metabolomics for a bioreactor application is to provide a fast and reliable way to estimate metabolite concentrations at different time stamps during a run of the bioreactor. Measured samples will likely have changing metabolite profiles throughout the run. This means that not only would the concentration of existing metabolites change but also new metabolites could appear when comparing experiments at different time stamps. Factors that promote a variable shift in resonance frequencies such as the pH and osmolality characteristics of the biofluid in the bioreactor will change over time. Solvent suppression NMR experiment protocols such as drying H2O are possible for a small number of samples, but they are difficult to implement in an online bioreactor application. This calls for a quantifier that is robust to deviations from the spectra of metabolites from databases, and is robust to the presence of a high-intensity NMR signal that is produced by the solvent.

An ideal quantifier for NMR metabolomics in this type of bioreactor application will provide fast and automated estimates for the absolute or relative concentrations of metabolites. The quantifier must be able to handle at least 40 metabolites in a cell growth media type of environment, and should operate directly from the NMR experiment data without too many manual interventions. Some of the challenges that make the development of such a quantifier difficult are: the small resonance frequency variations across experiments collected at different time stamps of a run of the bioreactor, the large dynamic range of the metabolite concentrations, the number of metabolites to quantify, the need for an automated approach, and the presence of different metabolites with nearby or overlapping resonance frequencies.

Of the three quantifiers, ASICS does not allow the user to specify a list of expected metabolites, rDolphin allows for the user to input any metabolites in the form of a custom profile, and Bayesil allows the user to specify a list of metabolites from their internal library via a website interface. rDolphin utilizes a profile system: physical chemistry parameters such as chemical shift values and J-coupling constants for each targeted metabolite are used in its quantification algorithm. rDolphin comes with three premade profiles: blood, fecal matter, and urine.

### 6.1. Data Preprocessing

We can see significant artifacts in the autophased data spectrum used by Bayesil in Figure 4, such as the presence of negative peaks in the 3–4 ppm region. The autophased data spectrum used by ASICS does not have this issue (see Figure 6). We recommend research efforts on improving the robustness of preprocessing algorithms for samples that have many overlapping metabolites, or on quantifiers that avoid preprocessing algorithms when possible.

### 6.2. Specificity and Sensitivity

The specificity of a quantifier is affected by the false positive detection and true negative detection of metabolites that are not in the test sample. In a bioreactor setting where NMR experiments are performed at regular time intervals, one usually has knowledge of the metabolites that are present at the beginning of the reaction. A quantifier that allows a user to override the list of metabolites that could be present in the test sample is likely to lead to better specificity.

Bayesil allows the user to choose a list of target metabolites, and we selected only metabolites that are known to be present in our DMEM sample. Neither false positive detection nor true negative detection is possible for Bayesil in our benchmark. ASICS, on the other hand, does not allow the user to manually select metabolites. The full list of metabolites reported to be in our data by ASICS is in the Appendix A. There are 20+ false positive detections; only 5 out of the 30+ reported metabolites match the DMEM composition, and these are listed in Table 4.

The sensitivity of a quantifier is affected by the true positive detection and false negative detection of metabolites that are in the test sample. Only a few of the metabolites in the DMEM composition were detected when using ASICS and rDolphin. ASICS detected several compounds that are similar to those in the DMEM composition, such as l-Alanine instead of l-Phenylalanine (see the Appendix A).

### 6.3. Uncertainty

Bayesil provides a confidence score for its estimates. Although some compounds estimated with high confidence still had large errors, e.g., l-Glutamine in Table 3, most of the lower confidence estimates indeed had large errors. The lower confidence seems to correlate with the low concentration level; see the specification’s relative concentration column S-RCG and S-RCL in Table 3. Bayesil was able to estimate the highly concentrated d-Glucose well when compared with the other quantifiers; see the REL column for the d-Glucose row in Table 3 against the same entry in other tables.

l-Leucine, l-Isoleucine, and l-Valine are metabolites that share a common spectral region for some of their high-intensity peaks, but have other less intense multiplet peaks in regions that are different from each other. Our sample composition (Table 2) shows that these three metabolites have similar concentrations. Bayesil had similar estimates for these three metabolites, as with ASICS and rDolphin-fecal for l-Leucine and l-Isoleucine; see the RCG column in Table 3, Table 4, Table 5 and Table 6.

### 6.4. Bias

Equation (Equation 9) implies that if the concentration of d-Glucose was underestimated, then the RCG of the other metabolites is likely to be overestimated. The errors of the rDolphin estimates from Table 5, Table 6 and Table 7 are mostly positive when relative concentrations are taken with respect to d-Glucose, and mostly negative when taken with respect to l-Leucine. The definition of error in Equation (Equation 10) implies a positive error indicates an overestimate, and a negative error indicates an underestimate. rDolphin overestimated d-Glucose and underestimated l-Leucine for our benchmark.

In the setting where experiments are collected at different time stamps during a run of the bioreactor, this bias behavior due to estimation error of metabolites suggests that relative concentration estimates should not be computed in reference to any compound that is changing between the experiments. This is because the particular estimation bias behavior might be inconsistent for different concentrations of the same compound. The NMR experiment 0 ppm reference compound (e.g., DSS) is chosen to be inert to the bioreactor reaction, and one can control how much of the reference compound is added to each experiment. We believe this control of the absolute concentration might help maintain consistency of a quantifier’s estimation bias on the reference compound across the experiments. Therefore, we suggest one should compute the relative concentration estimates in reference to the NMR reference compound.

In our benchmark, we used d-Glucose and l-Leucine as relative concentration reference because we were unable to get an estimate of DSS, the reference compound used in our benchmark test sample, from some of the quantifiers reviewed. This highlights the need for being able to estimate the concentration of common NMR experiment reference compounds that are used in metabolomics.

## 7. Conclusions

In this review, we discussed the challenges of building an automatic metabolite concentration quantifier algorithm for 1D ${}^{1}$H NMR experiments. The main challenges are the low sensitivity of NMR spectroscopy, variable peak shift due to experimental conditions, the overlapping of spectra from different metabolites, and the presence of the phase parameter for every resonance component in the data. We reviewed the FID observation model, which is a generative model for the data collected by 1D ${}^{1}$H NMR experiments. Popular preprocessing methods were also discussed. We surveyed the existing quantifiers that handle this type of NMR experiments, and benchmarked the Bayesil, the ASICS, and the rDolphin quantification software on a 1D ${}^{1}$H NMR experiment for a commercial cell growth medium that is generally representative of a bioreactor sample.

All three methods produced quantification results within a few minutes, but there were significant errors for some metabolites. These quantifiers were not designed to handle the quantification challenges for cell culture bioreactor applications, and we hope our review provides some insights that can aid the development of future quantifiers that focus on the bioprocessing domain. For quantifying bioreactor experiments, we advise using manual quantification if precision is important. Since computational speed is not an issue for these quantifiers, we hope future quantification research efforts could afford to explore approaches that have more computational burden but reduced user involvement.

A future direction of interest may be to incorporate more NMR theory to supplement the FID observation model. There have been efforts to standardize NMR-related data to include physical chemistry parameters for each metabolite [50]. It may be worthwhile to investigate how to utilize these physical chemistry parameters obtained from empirical fitting against simulated NMR spectra for the NMR metabolite quantification problem.

Another promising direction is to use a data-driven approach to predict the resonance frequency shifts due to varying experimental conditions. Studies such as [34] hint that for some metabolites in urine samples, accurate predictive mapping is possible. This kind of study requires a large number of NMR experiments with known concentration of the metabolites, e.g., 3000+ NMR experiments that are composed of random metabolite concentrations were acquired in [34]. Due to the relatively low cost of 1D ${}^{1}$H NMR experiments, acquiring a large number of NMR experiments to improve quantification performance or lower the quantifier’s computational resource requirements might be financially feasible for important high-throughput bioreactor applications.

## Figures and Tables

**Figure 1 metabolites-11-00157-f001:**
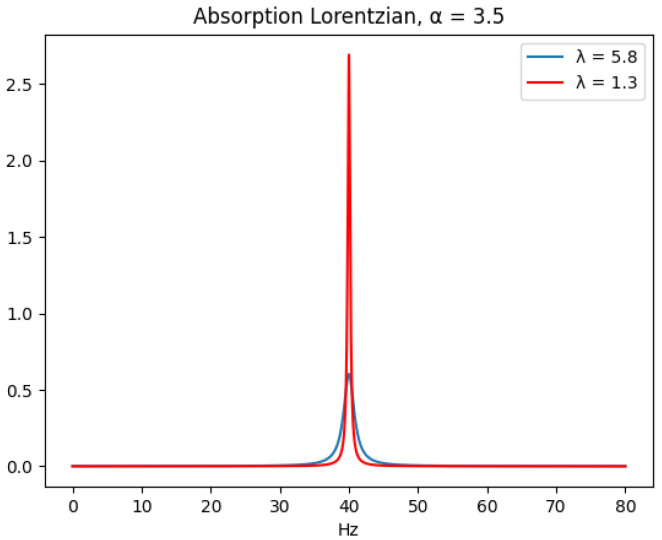
Two absorption Lorentzians that share the same frequency parameter. Both are multiplied by an intensity parameter α, which is set at 3.5.

**Figure 2 metabolites-11-00157-f002:**
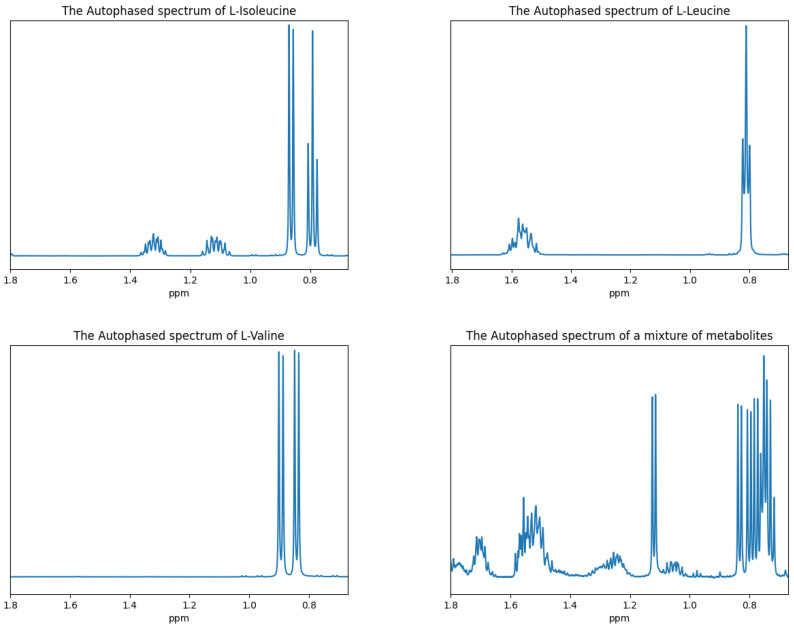
The autophased spectrum of four NMR experiments: (**top left**) experiment containing only l-Isoleucine, (**top right**) experiment containing only l-Leucine, (**bottom left**) experiment containing only l-Valine, and (**bottom right**) experiment containing a mixture of l-Isoleucine, l-Leucine, l-Valine, and other metabolites. Autophasing is discussed in Section 3.5.

**Figure 3 metabolites-11-00157-f003:**
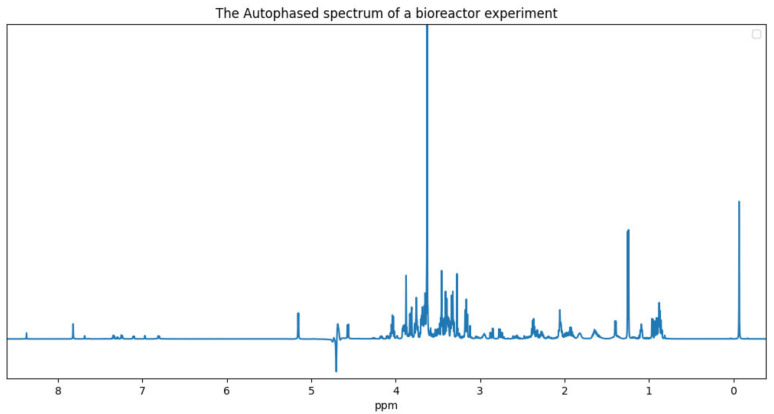
The real part of the autophased spectrum of an mammalian cell bioreactor NMR experiment.

**Figure 4 metabolites-11-00157-f004:**
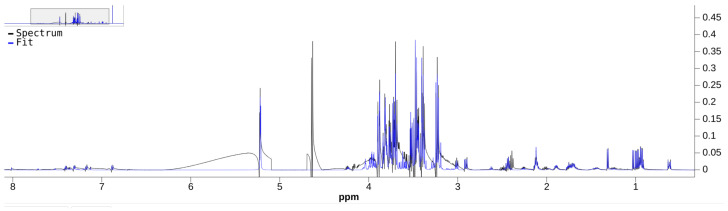
Reconstructed spectrum (blue, label: fit) vs. the preprocessed data spectrum (green, label: spectrum). Some significant artifacts are visible in the preprocessed data spectrum.

**Figure 5 metabolites-11-00157-f005:**
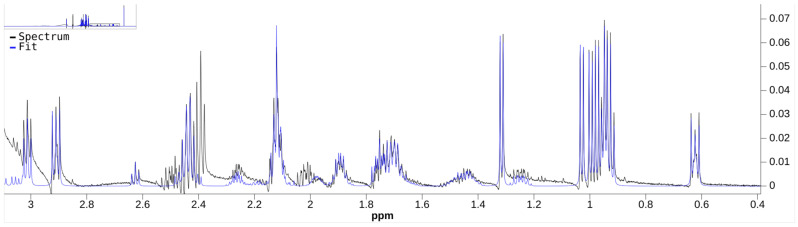
Close-up of Figure 4.

**Figure 6 metabolites-11-00157-f006:**
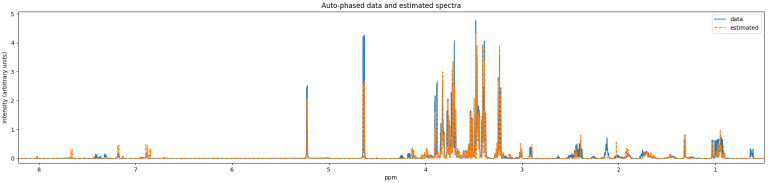
Reconstructed spectrum (orange, label: estimated) vs. the preprocessed data spectrum (blue, label: data).

**Figure 7 metabolites-11-00157-f007:**
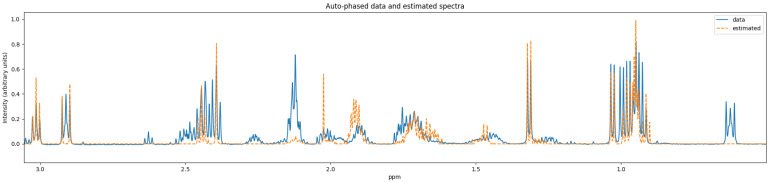
Close-up of Figure 6.

**Table 1 metabolites-11-00157-t001:** Comparison between biofluid vs. cell culture NMR experiments.

	Bioreactor/Cell Cultures	Biofluids
Continual measurement	Possible	Impossible
Sample size	Nonlimiting	Limiting
Condition changes	Straightforward	Difficult
Condition control	Straightforward	Possibly difficult
Sample preprocessing	Minimal	Possible
Number of samples	Possible in 1000’s	Mostly below 100
2D NMR or additional MS experiments	Possible for small subset of samples	Possible
Sample collection effort	Trivial	Complicated
Dynamic range across metabolites and samples	Very large across metabolites and samples	Large in some samples (e.g., urine)

**Table 2 metabolites-11-00157-t002:** Absolute concentrations from the Dulbecco’s modified Eagle’s medium (DMEM) sample.

Compound	Concentration (mg/L)
Glycine	30
l-Arginine	84
l-Cystine	62.57
l-Glutamine	584
l-Histidine	42
l-Isoleucine	105
l-Leucine	105
l-Lysine	146
l-Methionine	30
l-Phenylalanine	66
l-Serine	42
l-Threonine	95
l-Tryptophan	16
l-Tyrosine	103.79
l-Valine	94
Choline	4
d-Ca-Pantothenate	4
Folic acid	4
Nicotinamide	4
Pyridoxal	4
Riboflavin	0.4
Thiamine	4
i-Inositol	7.2
d-Glucose	4500

**Table 3 metabolites-11-00157-t003:** Estimated relative concentration of metabolite from Bayesil. Prefix *S-* stands for the DMEM product specification, prefix *B-* stands for Bayesil, and RCG stands for relative concentration compared to d-Glucose. REG is the relative error for RCG, and AEG is the additive error for RCG. The heading RCL, REL, and AEL are similarly defined for concentration relative to l-Leucine. C stands for confidence score: higher is more confident. It ranges between 1 to 10, with 10 being most confidence. All numeric values are rounded to three significant figures.

Compound	S-RCG	B-RCG	REG	AEG	S-RCL	B- RCL	REL	AEL	C
Glycine	0.00667	0.00969	0.453	0.00302	0.286	0.362	0.267	0.0762	10
l-Arginine	0.0187	0.012	−0.357	−0.00666	0.8	0.448	−0.439	−0.352	6
l-Glutamine	0.13	0.228	0.759	0.0985	5.56	8.53	0.533	2.96	10
l-Histidine	0.00933	0.0434	3.65	0.0341	0.4	1.62	3.05	1.22	7
l-Isoleucine	0.0233	0.0343	0.468	0.0109	1	1.28	0.28	0.28	7
l-Leucine	0.0233	0.0268	0.148	0.00344	-	-	-	-	10
l-Lysine	0.0324	0.0327	0.00889	0.000289	1.39	1.22	−0.121	−0.168	9
l-Methionine	0.00667	0.0108	0.613	0.00408	0.286	0.402	0.405	0.116	10
l-Phenylalanine	0.0147	0.0172	0.173	0.00254	0.629	0.643	0.0226	0.0142	10
l-Serine	0.00933	0.101	9.87	0.0922	0.4	3.79	8.48	3.39	4
l-Threonine	0.0211	0.035	0.66	0.0139	0.905	1.31	0.447	0.404	9
l-Tryptophan	0.00356	0.00985	1.77	0.00629	0.152	0.368	1.41	0.215	5
l-Tyrosine	0.0231	0.00542	−0.765	−0.0176	0.988	0.202	−0.795	−0.786	10
l-Valine	0.0209	0.0335	0.601	0.0126	0.895	1.25	0.396	0.354	10
Choline	0.000889	0.00452	4.08	0.00363	0.0381	0.169	3.43	0.131	10
i-Inositol	0.0016	0.0427	25.7	0.0411	0.0686	1.59	22.2	1.52	9
d-Glucose	-	-	-	-	42.9	37.3	−0.129	−5.51	10

**Table 4 metabolites-11-00157-t004:** Results from the ASICS quantifier. The column headings are defined for this quantifier (with prefix *A-*) analogous to the caption of Table 3, except RCG stands for relative concentration compared to d-Glucose-6-Phosphate.

Compound	S-RCG	A-RCG	REG	AEG	S-RCL	A-RCL	REL	AEL
Glycine	0.00667	0.164	23.6	0.158	0.286	0.34	0.191	0.0546
l-Isoleucine	0.0233	0.346	13.8	0.323	1	0.718	−0.282	−0.282
l-Leucine	0.0233	0.483	19.7	0.459	-	-	-	-
l-Cystine	0.0139	0.539	37.8	0.525	0.596	1.12	0.876	0.522
d-Glucose/d-Glucose-6-Phosphate	-	-	-	-	42.9	2.07	−0.952	−40.8

**Table 5 metabolites-11-00157-t005:** Results from the rDolphin quantifier with the blood profile. The column headings are defined for this quantifier (with prefix *rDb-*) analogous to the caption of Table 3.

Compound	S-RCG	rDb-RCG	REG	AEG	S-RCL	rDb-RCL	REL	AEL
l-Isoleucine	0.0233	0.447	18.1	0.423	1	0.673	−0.327	−0.327
l-Leucine	0.0233	0.663	27.4	0.64	-	-	-	-
l-Valine	0.0209	0.332	14.9	0.311	0.895	0.501	−0.441	−0.395
l-Glutamine	0.13	1.35	9.37	1.22	5.56	2.03	−0.635	−3.53
l-Lysine	0.0324	0.386	10.9	0.354	1.39	0.582	−0.582	−0.809
l-Methionine	0.00667	0.058	7.7	0.0513	0.286	0.0874	−0.694	−0.198
Glycine	0.00667	0.0229	2.43	0.0162	0.286	0.0345	−0.879	−0.251
l-Threonine	0.0211	0.147	5.98	0.126	0.905	0.222	−0.755	−0.683
l-Tyrosine	0.0231	0.0991	3.3	0.0761	0.988	0.149	−0.849	−0.839
l-Phenylalanine	0.0147	0.0882	5.02	0.0736	0.629	0.133	−0.788	−0.496
d-Glucose	-	-	-	-	42.9	1.51	−0.965	−41.3

**Table 6 metabolites-11-00157-t006:** Results from the rDolphin quantifier with the fecal profile. The column headings are defined for this quantifier (with prefix *rDf-*) analogous to the caption of Table 3.

Compound	S-RCG	rDf-RCG	REG	AEG	S-RCL	rDf-RCL	REL	AEL
l-Isoleucine	0.0233	0.86	35.9	0.837	1	0.632	−0.368	−0.368
l-Leucine	0.0233	1.36	57.3	1.34	-	-	-	-
l-Valine	0.0209	0.974	45.6	0.954	0.895	0.716	−0.2	−0.179
l-Lysine	0.0324	2.89	88.2	2.86	1.39	2.13	0.53	0.737
l-Methionine	0.00667	0.104	14.6	0.0972	0.286	0.0763	−0.733	−0.209
l-Tyrosine	0.0231	0.385	15.7	0.362	0.988	0.283	−0.714	−0.706
d-Glucose	-	-	-	-	42.9	0.735	−0.983	−42.1

**Table 7 metabolites-11-00157-t007:** Results from the rDolphin quantifier with the urine profile. The column headings are defined for this quantifier (with prefix *rDu-*) analogous to the caption of Table 3.

Compound	S-RCG	rDu-RCG	REG	AEG	S-RCL	rDu-RCL	REL	AEL
l-Isoleucine	0.0233	0.447	18.1	0.423	1	0.673	−0.327	−0.327
l-Leucine	0.0233	0.663	27.4	0.64	-	-	-	-
l-Valine	0.0209	0.332	14.9	0.311	0.895	0.501	−0.441	−0.395
l-Glutamine	0.13	1.35	9.37	1.22	5.56	2.03	−0.635	−3.53
l-Lysine	0.0324	0.386	10.9	0.354	1.39	0.582	−0.582	−0.809
l-Methionine	0.00667	0.058	7.7	0.0513	0.286	0.0874	−0.694	−0.198
Glycine	0.00667	0.0229	2.43	0.0162	0.286	0.0345	−0.879	−0.251
l-Threonine	0.0211	0.147	5.98	0.126	0.905	0.222	−0.755	−0.683
d-Glucose	-	-	-	-	42.9	1.51	−0.965	−41.3
l-Tyrosine	0.0231	0.0991	3.3	0.0761	0.988	0.149	−0.849	−0.839
l-Phenylalanine	0.0147	0.0882	5.02	0.0736	0.629	0.133	−0.788	−0.496

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
