# Peer review of "Automatic 1D ^1^H NMR Metabolite Quantification for Bioreactor Monitoring"

_metabolites, 2021, doi:10.3390/metabo11030157_

Round 1
Reviewer 1 Report
GENERAL COMMENTS:
The review from Wang et al. provides detailed information on NMR metabolite quantification, with focus on bioreactor applications. In the manuscript, also in-house data is used to show the differences in quantification using three different methods.
Some checking in the English language and slight rephrasing is necessary, but overall the manuscript is well-written.
In order to gain more usefulness to the reader, though, more information and examples related to bioreactors are needed, especially due to the focus of the review. Suggestions on the specifics of these additional parts are reported below.
SPECIFIC COMMENTS:
ABSTRACT
Line 4: NMR metabolomics is “a” method
Line 8: remove “the” before high-throughput
INTRODUCTION
NMR is never defined before line 22. Should be defined the first time it is mentioned.
Line 19: was it throughput instead of throughout? I would rephrase this last bit “possibly in high throughput”, might be unclear to the reader.
Line 27: remove extra space before the comma.
Line 37: maybe best to use “their signal(s)”
Line 38: add the first author of the review in text.
Line 42: “take” not “takes”.
Line 46: “remain” not “remains”.
Line 71-74: Rephrase a little the first part, not too clear to the reader.
1D 1H NMR PRELIMINARIES
Line 111: “has” instead of “have”.
Line 116: “weights” instead of “weight”.
Line 145: remove “the” before “an”.
CHALLENGES FOR AUTOMATIC QUANTIFICATION FOR BIOREACTOR MONITORING
Line 177: “has” not “have”.
Line 179: “selected” instead of “select”.
Figure 2: is it a choice to show unphased spectra?
Line 187: “a significant concentration difference”.
Line 189: “in quantification” or use “any uncertain quantification”
Line 192: “A” not “An”.
Line 211-213: I would add more lines on proposed substitutes for DSS. That is what is most interesting for a reader.
A small paragraph illustrating the main possible solvents used in bioreactors and how they are generally employed in NMR spectroscopy would be useful for the reader.
EXISTING QUANTIFIERS
Lines 273-274: missing a part of the sentence or rephrase.
Line 341: TSP was never defined before. Please add full name.
Line 362: report the full name of LASSO acronym at first.
Line 359: “normalizes”.
Line 363: “removes”.
BENCHMARK RESULTS
More experimental details on the NMR experiment used should be provided, maybe in Supplementary Material.
Line 388: “a 1D”.
Line 419: “has” not “does has”.
Table 2: some explaining on the confidence score would be useful to the reader.
Table 2: in the Supplementary i-inositol is then called myo-inositol. Stick to one naming.
DISCUSSION
Line 466: correct Bayesil.
Line 467: “utilizes”.
Line 482: maybe best to write “and these are listed in Table 3”.
Line 498-499: rephrase.
Line 509: “suggests”.
Author Response
We thank the reviewer for their careful review of the manuscript and their constructive suggestions. These helped us improve the quality of our manuscript.
General comment 1: Some checking in the English language and slight rephrasing is necessary,
Response: We have corrected all of the typographic errors that were highlighted in the reviewer's comments and completed another proofreading of the manuscript.
General comment 2: In order to gain more usefulness to the reader, though, more information and examples related to bioreactors are needed, especially due to the focus of the review.
Response: At the reviewer's suggestion, we decided to include the in-house data in the supplementary material. We hope this data can serve as a benchmark for the quantitative NMR community when they develop future quantifiers. These files include all instrumentation metadata for the exact settings used in the DMEM data that was used. Additionally we have added further information about the experimental protocol and sample in the text. In addition we have included a table (Table 1) that compares generic Bioreactor applications and generic applications of metabolomics in biofluids.
Specific comment 1: Figure 2: is it a choice to show unphased spectra?
Response: We have replaced the subfigures in Figure 2 with their auto-phased counterparts.
Specific comment 2: I would add more lines on proposed substitutes for DSS. That is what is most interesting for a reader. A small paragraph illustrating the main possible solvents used in bioreactors and how they are generally employed in NMR spectroscopy would be useful for the reader.
Response: Following Reviewer's suggestion, we have added further information about the internal references used in the case of NMR metabolomics. Concerning internal references, generic bioreactor applications are not any different than for other applications. In terms of solvent, our primary interest in this publication is in mammalian cell bioreactor for applications in gene therapy and vaccines production. We have also added a paragraph to Section 3.4 different experimental conditions (around line 223) .
Specific comment 3: EXISTING QUANTIFIERS, Lines 273-274: missing a part of the sentence or rephrase.
Response: The text referenced by the reviewer's comment is about time-domain quantifiers. Following the reviewer's suggestion, we have replaced our discussion on time-domain quantifiers with two paragraphs. In the revised manuscript, this can be found as the second and third paragraph of Section 4 Existing quantifier, around lines 293-308.
Specific comment 4: More experimental details on the NMR experiment used should be provided, maybe in Supplementary Material.
Response: Following the reviewer's suggestion, we have now provided the in-house data for this experiment, along with the acquisition metadata as logged by the Bruker spectrometer. Also, we have added all necessary information about the experiment at the beginning of “Section 5 Benchmarking Results“.
Specific comment 5: Table 2: some explaining on the confidence score would be useful to the reader.
Response: Note that Table 2 in the first submission is now Table 3 in the revised submission. Following the reviewer's suggestion, we added the following text to that table's caption:
"C stands for confidence score: higher is more confident. It ranges between 1 to 10, with 10 being most confidence".
This revised table is in Section 5.1 Bayesil benchmark. We have a sentence describing Bayesil’s confidence score in the second paragraph of Section 4.1 Bayesil (around line 350). Unfortunately, not much more was described in the Bayesil publication on these confidence scores.
Specific comment 6: Table 2: in the Supplementary i-inositol is then called myo-inositol. Stick to one naming.
Response: Following the reviewer's suggestion, we have changed all instances of myo-inositol to i-inositol in the supplementary document. Note it is now renamed to quantifier_outputs.xls.
Reviewer 2 Report
Comments to Manuscript by Chung Wang et al
“Automatic 1D 1H NMR metabolite quantification for bioreactor monitoring”
General comment
The authors performed a review of the tools available to performed automatic metabolite quantification in 1H-NMR spectra, focus on monitorisation of bioreactors. This is it is a current, important, and interesting topic. To my knowledge, this review is the first one focus in automatic metabolite quantification in bioreactor context.
The review is well written and meets the main points of the topic. The review in this manuscript in the current form is suitable for publication on Metabolites with some minor improvements.
Comments
- At least an automatic quantifier that met the authors criteria is not mention in the manuscript, Batman (Hao, J., Liebeke, M., Astle, W. et al. Bayesian deconvolution and quantification of metabolites in complex 1D NMR spectra using BATMAN. Nat Protoc 9, 1416–1427 (2014). https://doi.org/10.1038/nprot.2014.090). There is a reason for this quantifier do not being mentioned?
- There is a parameter that in my opinion is not well discussed in the manuscript: the baseline. Problems with the spectra baseline are very important for a correct metabolite quantification. Should be more discussed in the manuscript.
- In my opinion the review will be improved if the authors present the spectra acquired for the DMEM with the visual automatic profile of the spectra obtained with the different quantifiers.
- In the text, I would prefer that the quantifier be referred by the name and not by the reference number, it will be easier for the reader followed the text (example: line 292 “. For example, the quantifier [42] requires the user to specify various peak location…”).
- Despite not be directly the focus of this review, I think that the manuscript it will gain if the authors described better the experimental set-ups of systems with on-line monitorization of bioreactor metabolites by NMR.
Author Response
We thank the reviewer for their careful review of the manuscript and their constructive suggestions. These helped us improve the quality of our manuscript.
Comment 1: At least an automatic quantifier that met the authors criteria is not mention in the manuscript, Batman (Hao, J., Liebeke, M., Astle, W. et al. Bayesian deconvolution and quantification of metabolites in complex 1D NMR spectra using BATMAN. Nat Protoc 9, 1416–1427 (2014). https://doi.org/10.1038/nprot.2014.090). There is a reason for this quantifier do not being mentioned?
Response: Thank you for pointing out that we neglected to name this method. Furthermore, we know now we made an error in citing this paper when discussing this method in the text. We meant to cite the BATMAN quantifier, but we used an older (2012) publication from the same group led by Ebbels, who was also a coauthor of the BATMAN publication. In our original submission, this incorrect reference was reference 42:
Astle W., De Iorio M., Richardson S., Stephens D. and Ebbels T. M., A Bayesian model of NMR spectra for the deconvolution and quantification of metabolites in complex biological mixtures. J Am Stat Assoc 107, 1259-1271 (2012).
We revised the manuscript to use the correct reference for the BATMAN method, which is the one suggested by the reviewer. The discussion text about BATMAN remains unchanged but now clearly cites the method and names it. This text is the second paragraph before "Section 4.1 Bayesil", around line 321.
In this work we only tested methods that are currently maintained, that provide details of the approach and that allow loading of the data set in the provided tool. The BATMAN quantifier was excluded from our benchmark because it is no longer maintained. In fact, we were unable to load the setup data set for BATMAN, and this unresolved bug was reported by others for several years at the official BATMAN R repository: https://r-forge.r-project.org/.
Comment 2: There is a parameter that in my opinion is not well discussed in the manuscript: the baseline. Problems with the spectra baseline are very important for a correct metabolite quantification. Should be more discussed in the manuscript.
Response: Thank you for highlighting this issue. Our stance on baseline artifacts is that it is a generic term used for describing deviations from the Lorentizan model of the data collected. This deviation could have a number of complicated origins, some due to the instrument hardware, some due to the experiment protocol used, and some due to the data pre-processing algorithms used. A detailed discussion on possible causes would distract the focus of highlighting the requirements of bioreactor applications.
To incorporate the spirit of your suggestion, we have added two paragraphs to the end of Section 3.5 (around line 257) to introduce the reader to this topic. There, we added the following three reference for existing baseline compensation algorithms:
Wang et. al. "Distribution-Based Classification Method for Baseline Correction of Metabolomic 1D Proton Nuclear Magnetic Resonance Spectra", 2013, doi.org/10.1021/ac303233c
Xi et. al. "Baseline Correction for NMR Spectroscopic Metabolomics Data Analysis", 2008, doi.org/10.1186/1471-2105-9-324
Golotvina et. al. "Improved Baseline Recognition and Modeling of FT NMR Spectra", 2000, doi.org/10.1006/jmre.2000.2121
In those two new paragraphs, we also added the following discussion article on possible sources of baseline artifacts, in addition to the informative reference by Tang (1994, doi.org/10.1006/jmra.1994.1160) that we already have.
Marion et. al, "Baseline distortion in real-fourier-transform NMR spectra", 1988, doi.org/10.1016/0022-2364(88)90230-2
Comment 3: In my opinion the review will be improved if the authors present the spectra acquired for the DMEM with the visual automatic profile of the spectra obtained with the different quantifiers.
Response: Following the reviewer's suggestion, we have added four plots to show the reconstructed spectra of the Bayesil and ASICS quantifier. We were unable to find an analoguous option for rDolphin. In case the community is interested exploring further, We included the material required to generate the plots for Bayesil and ASICS in the supplementary material.
The new figures are Figures 4-7, and are added at the end of sections "5.1 Bayesil benchmark" and "5.2 ASICS benchmark". At the end of section “5.3 rDolphin benchmark”, we added the sentence "Unfortunately, We were unable to find an option in rDolphin to visualize the reconstructed spectra of all the metabolites on the same plot.".
In section "6 Discussion", we added a new unnumbered subsection heading "Data pre-processing". This subsection is a brief discussion on the four new figures. Although we discuss the suitability of each quantifier to the challenges we laid out for the monitoring application, we avoid a thorough systematic comparison between the quantifiers benchmarked in this review. Therefore we decided not to discuss in detail the particular data pre-processing steps used in their quantifier. The goal of our benchmark is to see whether any existing, publicly available, and still maintained quantifiers are ready for use with online monitoring applications.
Comment 4: In the text, I would prefer that the quantifier be referred by the name and not by the reference number, it will be easier for the reader followed the text (example: line 292 “. For example, the quantifier [42] requires the user to specify various peak location…”).
Response: We have revised the entire section "4 existing quantifiers" to include the method name in addition to the citation, as per the reviewer's excellent suggestion. When there is no method name, we resort to using the first author of the method as the method name.
Comment 5: Despite not be directly the focus of this review, I think that the manuscript it will gain if the authors described better the experimental set-ups of systems with on-line monitorization of bioreactor metabolites by NMR.
Response: Thank you for this suggestion. We have added a sentence on the flow NMR probe as a way to perform on-line monitoring. This is in the second paragraph of the revised manuscript, around line 30. As we do not aim to promote any manufacturer in this review we did not list specific product. Also, we have added a Table 1 to the beginning of "Section 3: Challenges for automatic quantification for bioreactor monitoring". We added a paragraph (first paragraph of Section 3) that discusses Table 1.
To encourage the community to develop and evaluate novel quantifiers, we have now provided the complete DMEM NMR data that was used in our manuscript as part of our supplementary files.
Reviewer 3 Report
The review by Wang et al. is well-written and informative. It lays out the need and challenges of automated on-line metabolite quantification of a bioreactor and demonstrates the inherent limitations of current quantifier software. Just a few minor points:
- Figure 2 appears distorted with likely phasing and shimming problems. This should be addressed and fixed.
- The three quantifier software have different limitations in the selection of metabolites, so the current comparison, while informative, is a little misleading. Can the calculations be repeated with a metabolite reference library (as best as possible) that is the same for all three software packages? In this regard, the relative performance of false positive rates, false negative rates, and quantitation accuracy can be directly compared.
It would also be valuable to report and compare an overall %CV in the quantification of the metabolites correctly detected relative to the DMEM values.
Author Response
We thank the reviewer for their constructive suggestions. These suggestions have helped us improve the quality of our manuscript.
Comment 1: Figure 2 appears distorted with likely phasing and shimming problems. This should be addressed and fixed.
Response: Thank you for pointing out that the figure appears distorted. It is because we used the magnitude spectrum. We have now replaced the subfigures in Figure 2 with their auto-phased versions.
Comment 2: The three quantifier software have different limitations in the selection of metabolites, so the current comparison, while informative, is a little misleading. Can the calculations be repeated with a metabolite reference library (as best as possible) that is the same for all three software packages? In this regard, the relative performance of false positive rates, false negative rates, and quantitation accuracy can be directly compared.
It would also be valuable to report and compare an overall percent-CV in the quantification of the metabolites correctly detected relative to the DMEM values.
Response: We speculate that the reviewer is asking us to use a common platform of comparison for the three quantifiers, concerning the sensitivity and specificity performance of each.
We think the reviewer is suggesting us to first find the list of common metabolites across the internal libraries of the three quantifiers. Then, attempt to force each quantifier to do targeted analysis for these common metabolites. This subjects each quantifier to the same targeted analysis, i.e., they all have to produce concentration estimates for the same types of metabolites. This offers a fair comparison of sensitivity and specificity of the detection of metabolite types across the quantifiers.
Unfortunately, we do not have access to the internal library of the current version of ASICS. Furthermore, We were unable to create a custom profile for rDolphin so that it would do targeted analysis on only metabolites in DMEM; doing so also required us to manually input all chemical shift and J-coupling constant values for each compound to be targeted into the rDolphin configuration file. The chemical shift values change according to pH and other experimental factors as discussed in our manuscript. The manual effort in tuning the configuration file for each experiment is a serious issue when developing an automated quantifier for online monitoring applications.
In conclusion, we agree that the reviewer's suggested experiment would permit an unbiased comparison for each quantifier against each other, but it would not reflect how NMR quantification will be used in a generic online monitoring setting. Our motivation for this review is to inform the community that existing quantifiers don’t seem to be doing well against a typical sample from an online monitoring setting. To assist the community in further diagnosing the issues with specific quantifiers and developing future quantifiers, we have now provided the complete DMEM NMR data that was used in our manuscript as part of our supplementary files.
Round 2
Reviewer 3 Report
The authors have adequately addressed my prior concerns.